# The Impact of Agricultural Drought on Smallholder Livestock Farmers: Empirical Evidence Insights from Northern Cape, South Africa

**Yonas T. Bahta** \* and **Vuyiseka A. Myeki**

Department of Agricultural Economics, University of the Free State, P.O. Box 339,
Bloemfontein 9300, South Africa; myekiasah@gmail.com
\* Correspondence: bahtay@ufs.ac.za; Tel.: +27-514-019-050

**Abstract:** The Northern Cape Province of South Africa faces drought that limits human endeavours, and which explains the unpredictable progress in livestock production over time. This study evaluated the impact of agricultural drought on smallholder farming households' resilience in the Northern Cape Province. Data from 217 smallholder livestock farmers were collected and analysed descriptively and with the Agricultural Drought Resilience Index (ADRI), and Household Food insecurity Access Scale (HFIAS). This study found that most smallholder livestock farmers (79%) were not resilient to agricultural drought. Further, the findings revealed that agricultural drought significantly impacted resources, food security, and government policy. This implies that a lack of access to resources, food insecurity, and ineffective government policy affected already vulnerable smallholder livestock farmers coping with severe agricultural drought. This study is significant in providing policymakers and other stakeholders with evidence-based recommendations for developing strategies and implementing policies for improving the resilience of smallholder livestock farmers by improving access to resources, including access to land. They will be better able to deal with challenges that come their way if they become more resilient, resulting in reduced household loss. This study recommends that government and stakeholders enhance the resilience of smallholder farmers by supporting less resilient farmers. The government needs to work with stakeholders to improve access to land and funds to enhance farmers' resilience. As a result, these policies can help smallholder farmers be more resilient in times of climatic shock.

**Keywords:** resources; resilience; food security; government policy; household food insecurity access scale; agricultural drought resilience index

## 1. Introduction

Climate change has a significant impact on food production around the world, with drought being one of the worst natural disasters that can occur, affecting the majority of the world's population [1]. Agricultural drought is the most costly natural disaster globally, causing severe damage and affecting more people than any other natural disaster [2,3]. Agricultural drought has ravaged countries all over the world, impacting harvests and the game, meat, milk, and fruit industries.

Existing international and national studies, such as those by Lottering et al. [4], Holman et al. [5], Algur et al. [6], Matlou et al. [7], Zwane [8], Mare et al. [9], and Chen et al. [10], have reviewed the impact of drought on smallholder farmers, drought impact and response, impact of drought on livelihoods, impact of drought on the welfare of smallholder live-stock farmers, the impact of climate change on agriculture and food security, the impact of drought on commercial farmers, and productivity.

Lottering et al. [4] used a systematic approach to search for published literature on the effect of drought on small-scale farmers in Sub-Saharan Africa from 2008 to 2018. They discovered that droughts have far-reaching consequences for a country's environment,

society, and economy. There is, however, a scarcity of reliable and comprehensive information on the effects of drought, necessitating research on such effects and how they can be predicted and mitigated. Improving drought preparedness and mitigation is a critical prerequisite to reducing the vulnerability of small-scale farmers and rural communities to drought impacts.

Holman et al. [5] investigated the effects of and responses to drought. They discovered that the majority of reported responses were on-farm, and that a range of measures were implemented across institutional scales and throughout the supply chain, indicating complex interactions within the food system. Drought responses were dominated by reactive and crisis-driven actions to cope with or improve recovery from drought, which contributed little to increased resilience against future droughts.

Algur et al. [6] reviewed the impact of drought on the health and livelihoods of women and children. They discovered that climate change is expected to have severe global consequences, some of which are already being felt. Droughts are expected to become more frequent and intense in some areas in the 21st century. This necessitates deliberate government interventions to mitigate the effects.

Matlou et al. [7] investigated the impact of agricultural drought resilience on the welfare of smallholder livestock farming households. They discovered that smallholder farmers who received drought relief assistance fared better. However, the welfare improvements varied across respondent and gender categories, with men experiencing greater welfare improvements than women. This study also showed that economic, social, human, and natural capitals all had a significant impact on the welfare of smallholder farmers. In addition, this study found that smallholder farmers had a moderate agricultural drought resilience index but low natural resilience capital.

Zwane [8] conducted a literature review to assess the impact of climate change on primary agriculture and food security. The author discovered that many dams had low water levels, resulting in lower crop yields, including grapes. Droughts have become a common occurrence, affecting both smallholder and commercial farmers. Livestock production decreased over time, with small stock, beef, and dairy industries bearing the brunt.

Mare et al. [9] assessed the impact of drought on commercial livestock producers, with a focus on drought adaptation strategies. They revealed that drought had a significant effect on the average herd size, livestock feeding, and sheep flock. The government did not provide any assistance to commercial livestock producers. During the drought, the majority of farmers had no preventive measures in place.

Chen et al. [10] investigated the impact of drought on the interannual variability of net primary productivity. They found a strong positive relationship between global average moisture availability and net primary productivity. The positive relationship was a composite of events across dry regions and the net primary productivity decline during and after intensive drought events in humid regions. Importantly, they found many areas globally that did not show a strong correlation between drought and net primary productivity.

To the authors' best knowledge, none of the studies assessed the effect of agricultural drought on smallholder livestock farmers with respect to resources (access to land, access to water, assets), food security (consumption, employment, credit, saving), and government policy (drought relief program). Previous studies [4–10] focused on the impact of drought and livelihood, welfare, productivity, and food security drought response. Therefore, this study evaluated the impact of agricultural drought on smallholder livestock farming households' resilience in the Northern Cape Province, South Africa. Thus, this study is innovative as it incorporates detailed components of resources (access to land, access to water, assets), food security (consumption, employment, credit, saving), and government policy (drought relief program). This study assessed the effect of agricultural drought on smallholder livestock farmers, and what was needed to enhance the resilience of smallholder farming households and enable them to revert to their living conditions prior to the

drought. The findings will aid policymakers to formulate appropriate policy interventions to sustain smallholder livestock farmers against the impact of drought, which threatens farm and household economy, food security, economic growth, human survival, and the living standards of farmers.

*The Impact of Agricultural Drought, Recovery, and the Government's Response*

Agricultural drought is the costliest natural disaster on the planet. Agricultural drought has a negative impact on the livelihoods of farmers and economies in the developing world, where an estimated 166 billion USD loss has been recorded from three-quarters of the global cropped area of 454 million hectares. One drought event reduces the average agricultural gross domestic product by 0.8% globally, with magnitudes varying by country [11]. Due to limited resources, Africa is the most vulnerable to adapting to the effects of agricultural drought. Agricultural drought causes 80% of the economic losses in developing countries and has a negative impact on agricultural sustainability by reducing social well-being through food security and decreasing economic resources [12]. Furthermore, Melketo et al. [13] and Bahta and Myeki [12] highlighted that resources, policies, and access to credit all have a significant impact on the resilience of smallholder farmers.

In most countries, agricultural production relies highly on weather conditions and water availability. As drought impacts livestock, it can result in poor productivity, decreased fertility, poor animal health, and a rise in livestock mortality. Many farms suffer from a variety of contagions, most notably lung contagions, as a result of the dusty environment caused by drought. Furthermore, a lack of adequate grazing causes a large number of livestock to be deficient in essential nutrients, resulting in low conception rates, retained afterbirths, poor quality colostrum production, and immune deficiencies [14].

In South Africa, agricultural output was 9.2% lower in the first half of 2019 than in 2018. The drought, which is being called the worst in 100 years, affects about 37% of South Africa's rural population [15]. Approximately 590,000 km$^2$ of land were severely affected by agricultural drought in areas utilized for livestock farming, severely harming pastureland and increasing livestock mortality in most South African provinces [16].

The Northern Cape Province of South Africa drought is one of the worst the province has experienced in more than a century and impacted about 10,000 farms with a carrying capacity of 166,000 big stock units, covering more than 5.8 million hectares [17]. Approximately 86% of the land is used for grazing, and in the Northern Cape, about 33.8 million hectares comprise agricultural land [18].

The impact of agricultural drought on livestock production is a significant physical stressor in temperate and humid regions, including South Africa [19]. Agricultural drought impacts livestock production and quality, which depends on several factors, such as intensity, recurrent agricultural drought, vulnerability, water stress, and socio-economic circumstances [20]. Smallholder livestock farmers face numerous threats to agricultural production and their wellbeing. These threats are related to climate change vulnerability and include social, economic, and environmental shocks, which impact farm economy, household economy, and government policy [21].

The length of time required for herd recovery is determined by the severity of the drought, the effect on breeding females, and the amount of rainfall during the interdrought period [22]. According to studies, it takes two to three years to recover from a severe drought. According to research conducted in southern Ethiopia, land degradation, increased human population density, smaller herd sizes, and the loss of key resources have made pastoralists increasingly vulnerable to repeated droughts and unable to fully recover in the periods between successive droughts [22]. Vetter et al. [23] highlighted that cattle farmers in the KwaZulu-Natal provinces of South Africa lost 43% of their herd, compared with 29% for goats, in the 2015–2016 agricultural drought season. Cattle numbers remained low three years after the drought, whereas goat numbers had recovered. Larger herds had lower mortality rates, implying that owners of larger herds had more resources to support their herds.

With anticipated pressures on water resources, and more intense and severe droughts, a paradigm shift is required. Drought-related "crisis management" that is poorly coordinated will no longer suffice [1]. A well-planned strategy aimed at mitigating the effects of drought is required. The adoption of national drought policies focused on risk reduction, supplemented by drought mitigation plans at various levels of government, will have significant ramifications across key sectors [24].

By promoting integrated water resource management, these policies help to achieve Sustainable Development Goal target 6 "ensure availability and sustainable management of water and sanitation for all". The vulnerability to future drought episodes can be significantly reduced, and communities' and even entire nations' coping capacity can be improved. A proactive approach to improving drought resilience is built on three key pillars: drought monitoring and early warning systems, vulnerability and risk assessment, and drought risk mitigation measures [24].

South Africa's government is committed to assisting emerging and small-scale farmers, but it is also constrained by severe budget constraints and bureaucratic red tape surrounding drought relief provided by the government for deprived farmers in drought-stricken areas. Effective drought reduction and recovery strategies are required to reduce the economic burden of drought losses. Support for drought relief and recovery should take into account the different capabilities and needs of small and large herd owners. The effects of high stocking rates and resource scarcity on mortality and herd growth were visible during the drought year but not in other years [25]. Most die-offs appear to have occurred in a short period of time after forage and water resources became scarce and difficult to access. Recognizing when this threshold is being approached would be beneficial for targeting strategic interventions. Rainfall patterns, forage availability, water availability, and animal condition could all be indicators of potential concern.

## 2. Conceptual Framework

This study used a framework developed by the authors (Figure 1). Three important variables were incorporated: farm economy (resources–access to land, access to water, and assets), household economy (food security variables—consumption, employment, credit, and saving), and government policy (drought relief program) in relation to agricultural drought and agricultural drought resilience, measured by ADRI. This framework highlighted the impact of agricultural drought on resource access; the impact of agricultural drought on food security variables—how the shock affects the consumption of food, employment, savings, and access to credit, and how the government responds to the impact of drought using different drought relief programs to enhance smallholder livestock resilience. A detailed description of the variables is depicted in Table 1.

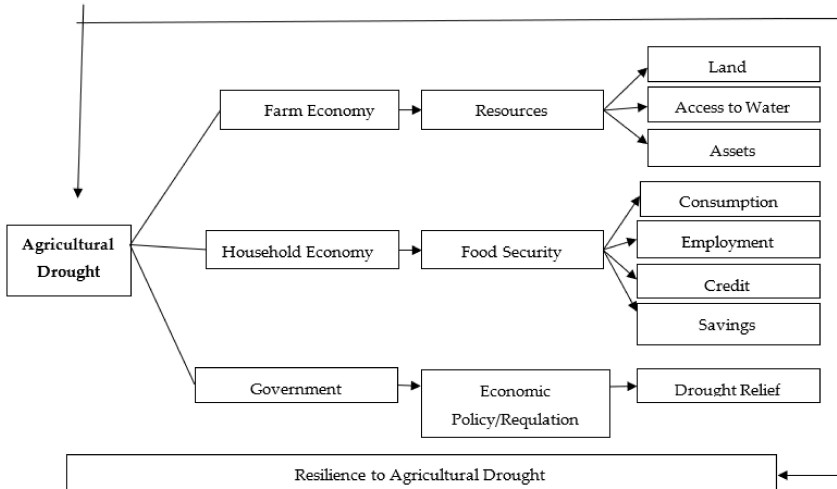

**Figure 1.** Conceptual framework. Source: Author's compilation.

**Table 1.** Description of variables.

| Variables | | Description |
|---|---|---|
| ADRI | Resilience/ no resilience | Impact on resilience to agricultural drought (ADRI > 0 = resilient, ADRI l < 0 = not resilient) |
| Farm Economy | Land | Impact on the availability of land for grazing and accommodating livestock, land ownership (customary = 1, rented = 2, purchased = 3, other = 4) |
| | Access to water | Impact on availability of water sources (dams, rivers): Does the household have any water sources (river/dam) close by? (Yes/No) |
| | Assets | Impact on the availability of household assets: Do you own any of the following assets: land, livestock, house etc.? (Yes/No) |
| Household Economy | Consumption | Impact on consumption: consumption in a normal year (a year without drought) vs. consumption in a drought year (how many Kgs does a household consume?) |
| | Employment | Impact on income generating sources (Farmer = 1, Employed = 2, Pension = 3, Unemployed = 4) |
| | Credit | Impact on support from a supporting financial institution: Do you have access to credit when you need it? (1 = Yes, 0 = No) |
| | Savings | Impact on saving: Do you have any savings to prepare and recover from drought? (Yes = 1, No = 0) |
| Government Policy/regulations | Drought relief | Impact on the policy implemented to enhance the resilience of smallholder livestock farmers: Do you get assistance from the government? (Yes = 1, No = 0) |

Between 2010 and 2013, significant effort was centred on developing conceptual frameworks of resilience that aid in understanding how shocks and stresses affect livelihood strategies and household well-being and in identifying key leverage points to be used in a theory of change, which in turn informs resilience-enhancing programming [26].

Several frameworks for resilience analysis have been proposed [27]. Hoddinott [28], on the other hand, contends that the plethora of frameworks for resilience analysis shares similar components. These include highlighting the broader environment in which a household resides, including the household and farm economy and the resources available to the households; how the shock affects food security; and how the government builds resilience through policies and interventions.

The economic viability of farming systems, or the ability to be profitable, is defined as economic wellness [29]. The household economy refers to a household's combined economic activities. The household economy is often referred to as the household sector instead of farm income, but the variables that contribute to household stability are often overlooked. The household sector is substantial enough to be called a household economy [30]. The government refers to economic policy or actions governments take to ensure business and farm development [31].

The conceptual framework elicits the extent of the impact of agricultural drought on smallholder farming households' resilience in the Northern Cape Province, South Africa. The conceptual framework is based on the factors that make farming households resilient

to agricultural droughts, such as farm economy, household economy, and government policy (Figure 1).

This selection of the framework is justified because it is mainly proposed for analysis (the impact of agricultural drought) and this framework elicits variation in the extent of resilience building from one household to the other and that this variation is determined by diverse factors. This includes the impact of agricultural drought on farm economy, including resources (land, access to water, and assets); impact on household economy, including food security variables (consumption, employment, credit, and saving); impact on government policies (drought relief); and impact on households' resilience to shocks, such as agricultural drought measured by the ADRI (Equations (1) and (2)). The ADRI was calculated using principal component analysis (PCA) and variables related to livestock production and consumption with and without a drought season (Table 2).

**Table 2.** Principal component analysis of the Agricultural Drought Resilience Index.

| Variables | Communalities | | Component Factors | Corr.ADRI |
|:---:|:---:|:---:|:---:|:---:|
| | Initial | Extraction | 1 | |
| $P_n$ | 1 | 0.935 | 0.967 | 0.894 |
| $P_d$ | 1 | 0.958 | 0.979 | 0.995 |
| $M_n$ | 1 | 0.280 | 0.963 | 0.890 |
| $M_d$ | 1 | 0.955 | 0.977 | 0.984 |

Total = 3.776. Eigenvalue's variances (%) = 94.402. Cumulative (%) = 94.402. KMO test of sampling adequacy = 0.636. Bartlett's test of sphericity is significant at $p = 0.0000$; chi-square = 2224.837.

Drought has a multiplier effect on various variables, including resilience, resources, food security, and policies. Household resilience is improved when there is sufficient access to resources—access to land, access to water, and access to assets. Household resilience is also improved through increased consumption, job creation, credit access, and saving. Drought relief policies, for example, have an impact on household resilience; if the government assists less resilient households in a timely manner, as it allows farmers to recover more quickly.

Smallholder livestock farmers rely heavily on natural resources, such as rainfall, which are directly impacted by climate change. Aside from a lack of rain, other factors contribute to smallholder farmers' vulnerability, such as frequent disasters, poverty, food insecurity, limited resources, limited adaptive capacity, weak infrastructure, and inadequate government policies, including limited drought relief, which does not arrive on time [12,32,33].

## 3. Materials and Methods

### 3.1. Study Area

The Northern Cape is the largest and most sparsely populated province of South Africa, and distances between towns are enormous due to its sparse population. Its capital is Kimberley. Its size is just shy of the size of the American state of Montana and slightly larger than that of Germany. The province is dominated by the Karoo Basin and consists mostly of sedimentary rocks and some dolerite intrusions. The south and south-east of the province is high-lying, 1200–1900 m, in the Roggeveld and Nuweveld districts. The west coast is dominated by the Namaqualand region, famous for its spring flowers. This area is hilly to mountainous and consists of granite and other metamorphic rocks. The central areas are generally flat with interspersed salt pans. Kimberlite (igneous rock) intrusions punctuate the Karoo rocks, giving the province its most precious natural resource, diamonds. The north is primarily Kalahari Desert, characterised by parallel red sand dunes and acacia tree dry savanna [34].

This study was conducted in the Northern Cape Province of South Africa (Figure 2). The Northern Cape Province is situated in the northwest region of South Africa and shares international borders with Botswana and Namibia and local borders with the Western and

Eastern Cape provinces in the south and the Free State and North West provinces in the east [35]. The province's land area is 372,889 km², accounting for 30.5% of South Africa's total land area, with a population of 1.2 million people [36]. Smallholder livestock farmers in the province use government-owned communal land for farming. They do not have access or full property rights because they do not own the land, and as a result, they have difficulty obtaining credit [13].

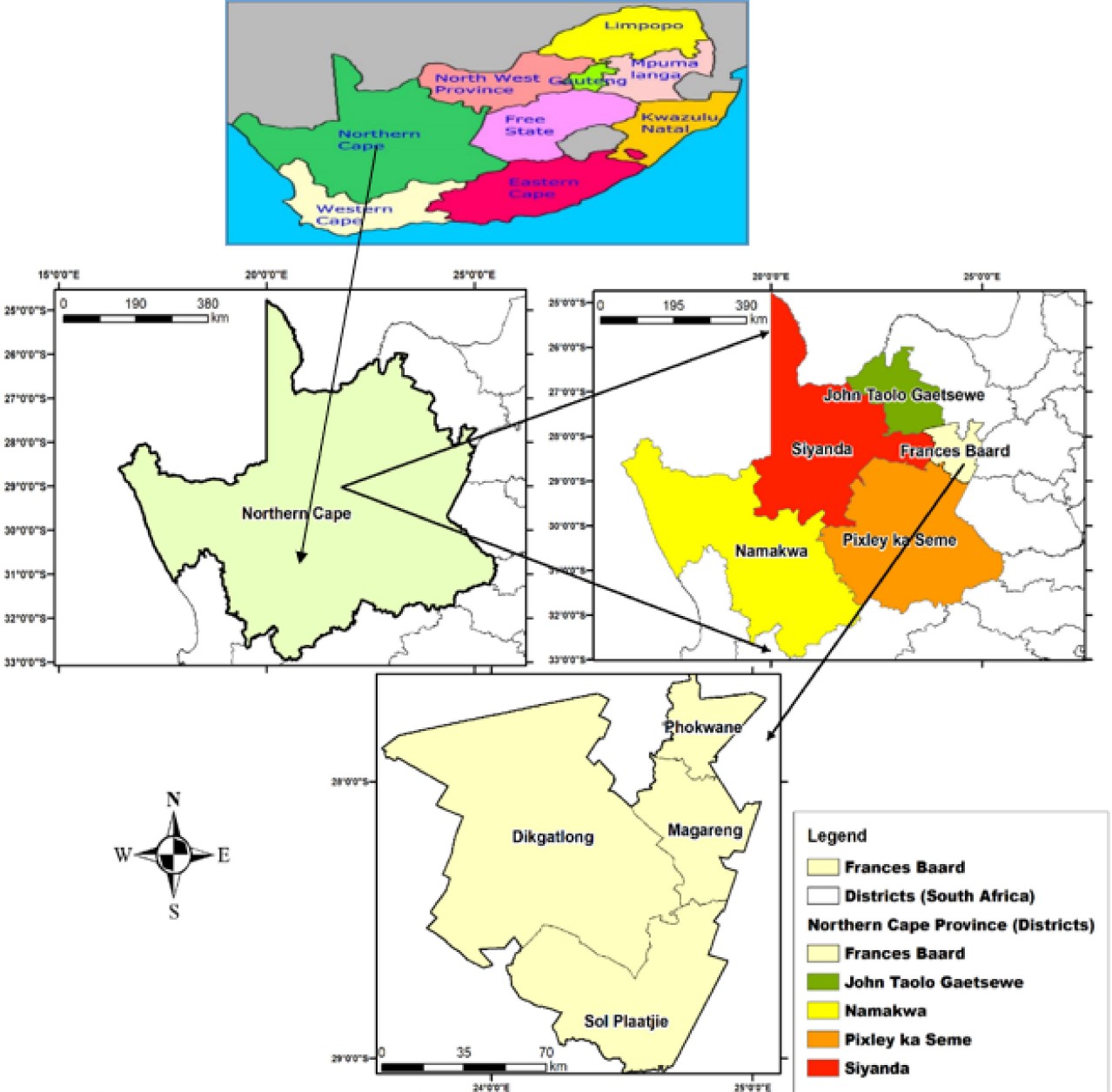

**Figure 2.** Maps of South Africa highlighting the Northern Cape Province, district municipalities of the Northern Cape, and the four local municipalities of FBDM. Source: FBDM [36].

The climate in the Northern Cape is arid and semi-arid. It is a large dry area with a wide range of temperatures and topographical features. Rainfall is infrequent, ranging from 50 to 400 mm per year. The provinces' average annual rainfall/precipitation is 202 mm, which varies from an average annual precipitation of 497 mm at Kimberly, average annual precipitation of 195 mm at Springbok, and average annual precipitation of 237 mm at Sutherland. Summer temperatures frequently rise above 40 °C. During winter, the average daytime temperatures are mild and may drop below 0 °C at night. Winter is usually

frosty, with the southern area sometimes becoming bitterly cold, which often receives snow, and the temperature occasionally drops below −10 °C [34,37]. Evaporation levels exceed the annual average rainfall, which varies from 66 mm at Port Nolloth on the west coast to 414 mm at Kimberley and 457 mm at Kuruman. The western areas, including Namaqualand, receive rainfall during the winter from April to September. The central, northern, and eastern parts of the province receive rain mostly in summer from October to February [34].

Figure 3 depicts a spatial distribution map of the study area's precipitation (FBDM). There is a clear rainfall gradient in the Northern Cape Province, as is the case with South Africa, with rainfall on the east and gradually decreasing as one moves to the west. The FBDM is located on the eastern corner of the Northern Cape Province, bordering the Free State and the Northwest Province. There is a noticeable rainfall gradient within the FBDM, with annual rainfall decreasing as one moves from north to south, with Phokwane, Magareng, and parts of the Dikgatlong local municipalities receiving higher rainfall compared to the Sol Plaatje Local municipality in the south [34,37].

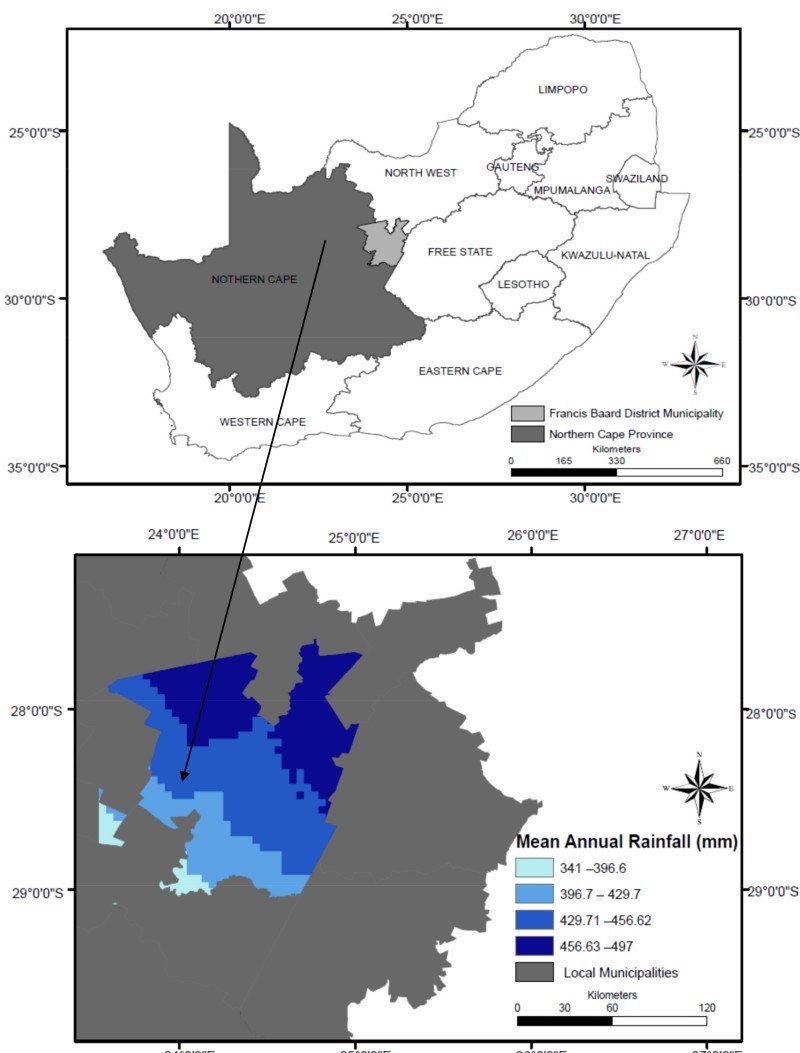

**Figure 3.** A spatial distribution map of the precipitation in FBDM. Source: Author's compilation and FBDM [36].

Frances Baard (12,800 km$^2$), John Taolo Gaetsewe (27,300 km$^2$), Namakwa (126,900 km$^2$), Pixley Ka Seme (103,500 km$^2$), and ZF Mgcawu (102,500 km$^2$) are the 5 district municipalities in the Northern Cape Province. This study was carried out in the Frances Baard District Municipality (FBDM), which is divided into four local municipalities: Dikgatlong

(2377.6 km$^2$), Magareng (1541.6 km$^2$), Phokwane (833.9 km$^2$), and Sol Plaatje (1877.1 km$^2$) (see Figure 2) [36]. The latitude and longitude of South Africa's Northern Cape Province are 29.0467° S and 21.8569° E, as shown in Figures 2 and 3. FBDM in Northern Cape Province is located at 29.0000° S and 25.0000° E.

Because of the vast differences in climate between the district municipalities, agricultural production in the Northern Cape Province of South Africa is diverse. Livestock production remains the most important enterprise, with 75% of agricultural households farming exclusively with animals [38]. In South Africa, the Northern Cape produces about 4% of the country's cattle, 24% of its sheep, 7% of its goats, and 1.4% of its chickens [39].

Despite the fact that the Northern Cape Province is accustomed to dry spells, the recent drought has crippled the agricultural sector, and recovery has been slow or non-existent. A lack of fodder and water for livestock production has placed an enormous strain on the industry [39]. The plight of smallholder farmers has worsened due to a variety of factors, such as poor grazing, a lack of water, a scarcity of resources, land disputes, and others. According to Matlou and Bahta [40], most smallholder farmers are not resilient to agricultural drought. Factors, such as a lack of credit, lack of government assistance (such as training and feed) during drought, lack of land ownership, and others, have contributed to their lack of resilience to agricultural drought.

The National Disaster Management Centre declared a drought disaster due to the persistent drought conditions in the Northern Cape and issued a notice in the Government Gazette of 2018 and 20 July 2021 classifying the drought in Northern Cape as a national disaster. The declaration was issued in terms of Section 23(1)(b) of the Disaster Management Act. Drought cycles are normal and natural; however, the drought conditions are regarded as the worst farmers have experienced in a century. Farmers from Northern Cape are battling the effects of the drought, the worst experienced in 100 years [41]. The Northern Cape has experienced an average of six to nine years of drought and received some relief through various state interventions. Further assistance is urgently needed to stabilise local economies and preserve what is left of core breeding herds [42].

According to the president of Agriculture Northern Cape, as cited by Coleman [43], livestock farmers in the Northern Cape lost entire herds and reduced their livestock numbers by more than 30% in the worst drought since 1982. The 2015/2016 year was the worst drought in South Africa, in general, and Northern Cape Province, in particular, putting a strain on the country's and province's agro-economic and water supply systems. The situation was exacerbated by insufficient drought relief schemes, inadequate policies, and a lack of disaster preparedness and collaboration amongst various stakeholders [9,44,45].

*3.2. Sampling Procedure*

This study followed a multistage sampling procedure. Firstly, the Northern Cape Province was selected purposefully from the rest of South Africa's provinces, because this province is a leading livestock producer. The province was also declared a disaster area by the South African government. The FBDM was chosen randomly in the second stage of the sample using balloting. Within FBDM, Dikgatlong, Sol Plaatje, Magareng, and Phokwane were purposely selected as the primary livestock-producing municipalities.

Finally, the sample frame was selected from a list of smallholder farmers who were identified and assisted throughout the 2015/2016 agricultural season. According to the Northern Cape Department of Agriculture, Forestry and Fisheries [46], the 4 local municipalities helped 878 smallholder livestock farmers register for assistance from the local government. The government assisted by giving animal feed and medication, expanding smallholder farmers' engagement in agricultural drought resilience measures by providing training and disseminating information, and enhancing access to agricultural financing and farm input. From the 878 farmers, 217 smallholder livestock producers were chosen using Cochran [47] and Bartlett et al.'s [48] simple random sampling procedure. A questionnaire was employed, with continuous and categorical data, such as the demographic and socio-cultural characteristics of smallholder livestock farmers. Face-to-face interviews

with smallholder livestock producers in the Northern Cape Province of South Africa were conducted from October to December 2020 using a structured questionnaire to acquire primary data. The questionnaire included both open-ended and closed-ended questions. The first section of the questionnaire asked for information on farmers' socio-economic characteristics, such as age, gender, household size, marital status, education, years of farming, and so on. The second section of the questionnaire sought information on the impact of drought on livestock, including impact on livestock health, price, and others. The third section of the questionnaire sought information needed to calculate the ADRI, which included production and consumption-related indicators in both normal and drought years (with agricultural drought year). The fourth section asked nine frequency of occurrence questions to assess food insecurity using the HFIAS. The final section sought information on government policy (drought relief policy), farm, and household economy-related variables. The University of the Free State provided ethical approval.

### 3.3. Data Analysis

### 3.3.1. Agricultural Drought Resilience Index (ADRI)

The ADRI was calculated using PCA to assess the resilience of smallholder farmers to agricultural drought. The ADRI was developed by aggregating four production and consumption-related indicators. The variables are the production of livestock in a normal year (without agricultural drought) (WnPn), livestock produced in a year with agricultural drought (WdPd), the period (number of months) during which the household consumed food produced in a year without drought (WcnMn), and the period (number of months) during which the household consumed food in a year with drought (WcdMd). The PCA reduces a large number of variables to smaller variables by considering the variance in the original data or variables [49,50]. The four indicators are aggregated into an ADRI (Equation (1)):

$$\text{ADRI} = W_n P_n + W_d P_d + W_{cn} M_n + W_{cd} M_d \tag{1}$$

where: ADRI: Agricultural Drought Resilience Index. W: The loading of components of the first principal weights determined. WnPn: Weight of livestock production in a normal year multiplied by the actual number of livestock produced. WdPd: Weight of livestock production in a normal year multiplied by the actual number of livestock produced. $W_{cn}M_n$: Weight for the number of the period (number of months) during which the household consumed food in a normal year multiplied by the actual number of food produced. WcdMd: Weight for the number of the period (number of months) during which the household consumed food in a drought year multiplied by the actual number of food produced.

### 3.3.2. Household Food Insecurity Access Scale (HFIAS)

The HFIAS, developed by the Food and Nutrition Technical Assistance Project (FANTA) [51], was applied to assess the impact of drought on food insecurity. The HFIAS score is a tool for measuring the degree of food insecurity in households in previous months. Therefore, each household's HFIAS score is calculated based on answers to nine 'frequency of occurrence' questions. The questions are:

- Concern about insufficient food—due to drought.
- Unable to consume preferred foods—due to drought.
- Consume limited kinds of foods—due to drought.
- Compelled to eat certain foods—due to drought.
- Eat lesser meals—due to drought.
- Eat fewer meals in a day—due to drought.
- No food of any kind in the household—due to drought.
- Go to sleep hungry—due to drought.
- Go a whole day and night without eating—due to drought.

The household's food security is determined by the score given to the above nine questions. The higher the score, the more food insecure the household is. Households are divided into four categories based on their HFIAS scores: food secure, mildly, moderately, and severely food insecure.

## 4. Results

### 4.1. Agricultural Drought Resilience Index (ADRI)

As presented in Table 2, due to the variables measuring the same construct, a high correlation among variables was observed. When considering the communalities and the initial communalities, it is clear that they are all greater than 0.30, which is a good sign. Based on the analysis of eigenvalues, one factor was extracted. The total variance explained indicates that 94.402% of the component explains the total variance. The results of the Bartlett test of sphericity are demonstrated. The results show that the null hypothesis is that the inter-correlation matrix is an identity matrix. The reduction of variables is rejected since the inter-correlation matrix did not derive from a population. The KMO statistics for the model amounted to 0.636, and the Bartlett test of sphericity was significant (*p*-value = 0.000 chi-square = 2224.837).

The ADRI can be written as (Equation (2)):

$$ADRI = P_n \times 0.967 + L_d P_d \times 0.979 + C_{cn} M_n \times 0.963 + C_{cd} M_d \times 0.977 \tag{2}$$

where: ADRI: Agricultural Drought Resilience Index. $P_n$: Production of livestock in a normal year. $P_d$: Production of livestock in a drought year. $M_n$: Number of the period (number of months) the household consumed food in a normal year. $M_d$: Number of the period (number of months) the household consumed food in a drought year. The numerical value: Weights derived using PCA (component factors).

### 4.2. Socio-Economic Characteristics of the Respondents

Table 3 depicts the socio-economic characteristics of the respondents. The average age of the farmers was 52 years. The average formal education of smallholder livestock farmers was eight years. The results show that 4% had a tertiary education, 42% a secondary education, and 54% of the respondents had a primary education. An average of 11 years of farming experience was observed. As indicated in Table 3, the minimum length of farming experience was half a year and the maximum was 60 years. The average number of household members was 5, with a minimum of 1 and a maximum of 25 members. From the study's findings, 61 (28%) of the respondents were women while 156 (72%) were men. The majority of the respondents were married (57%), followed by single (27%), widowed (9%), divorced (2%), and separated (1%), while the remaining respondents (4%) noted other (Table 3).

**Table 3.** Socio-economic characteristics of the respondents (*n* = 217).

| Indicators | | Occurrence | Percentage | Average | Min | Max | St.dev |
|---|---|---|---|---|---|---|---|
| Age (years) | 21–50 | 102 | 47 | 51.66 | 21.00 | 85.00 | 14.16 |
| | 51–85 | 115 | 53 | | | | |
| Education | Primary | 118 | 54.38 | 8.01 | 0.00 | 16.00 | 4.31 |
| | Secondary | 91 | 41.94 | | | | |
| | Tertiary | 8 | 3.68 | | | | |
| Farming experience (years) | 0.5–20 | 196 | 90.32 | 10.96 | 0.50 | 60.00 | 8.85 |
| | 21–60 | 21 | 9.68 | | | | |
| Household member | 1–10 | 204 | 94 | 5.19 | 1.00 | 25.00 | 2.88 |
| | 11–25 | 13 | 6 | | | | |
| Gender | Female | 61 | 28.1 | 0.72 | 0.00 | 1.00 | 0.45 |
| | Male | 156 | 71.9 | | | | |
| | Others | 9 | 4.1 | | | | |
| | Separated | 2 | 0.9 | | | | |
| Relationship status (Marital) | Divorced | 4 | 1.8 | 2.05 | 1.00 | 6.00 | 1.09 |
| | Widow | 19 | 8.8 | | | | |
| | Married | 123 | 56.7 | | | | |
| | Single | 59 | 27.2 | | | | |

Source: Author's compilation based on a survey.

### 4.3. Impact on Livestock

Agricultural drought has a significant impact on livestock production because it disrupts plant growth for animal feed, causing livestock growth to be delayed as a result. Prolonged periods of agricultural drought have been identified as one of the major constraints on South Africa's increasing demand for meat consumption being met. Therefore, agricultural drought came with a cost and caused significant livestock loss, with cattle accounting for 45%, sheep accounting for 46%, and goats accounting for 48%. Drought also had a negative impact on animal health, with 28% of cattle, 29% of sheep, and 27% of goats suffering a decline in health. Furthermore, drought led to a decline in livestock prices of 25% for cattle, 19% for sheep, and 19% for goats (Figure 4).

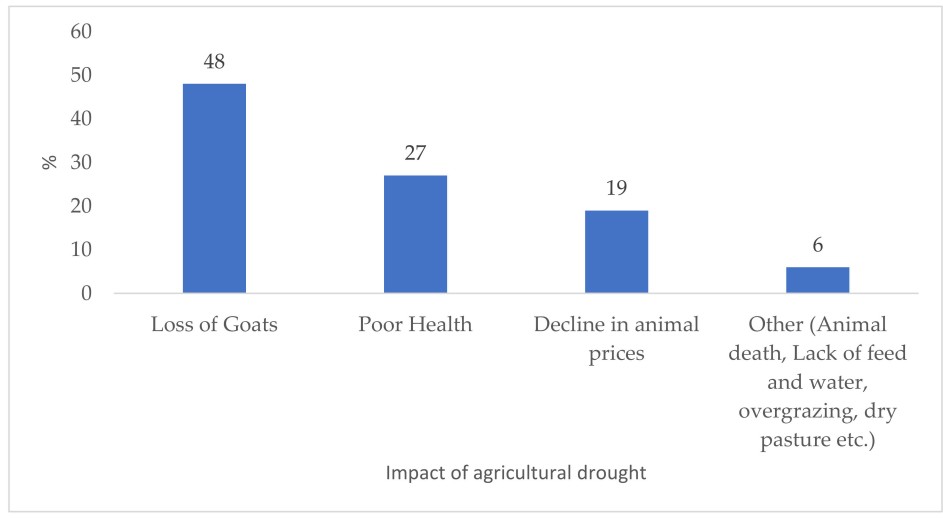

(**a**)

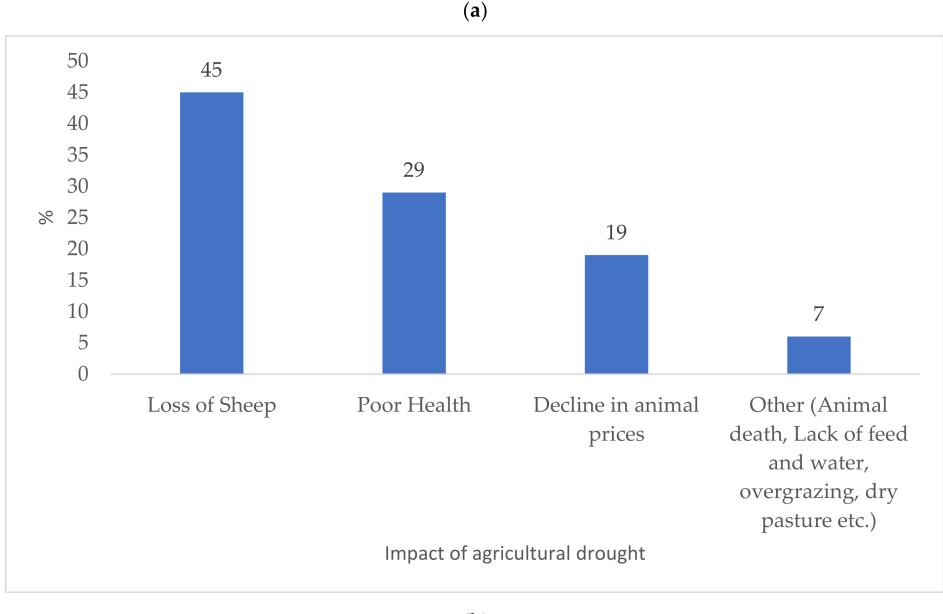

(**b**)

**Figure 4.** *Cont.*

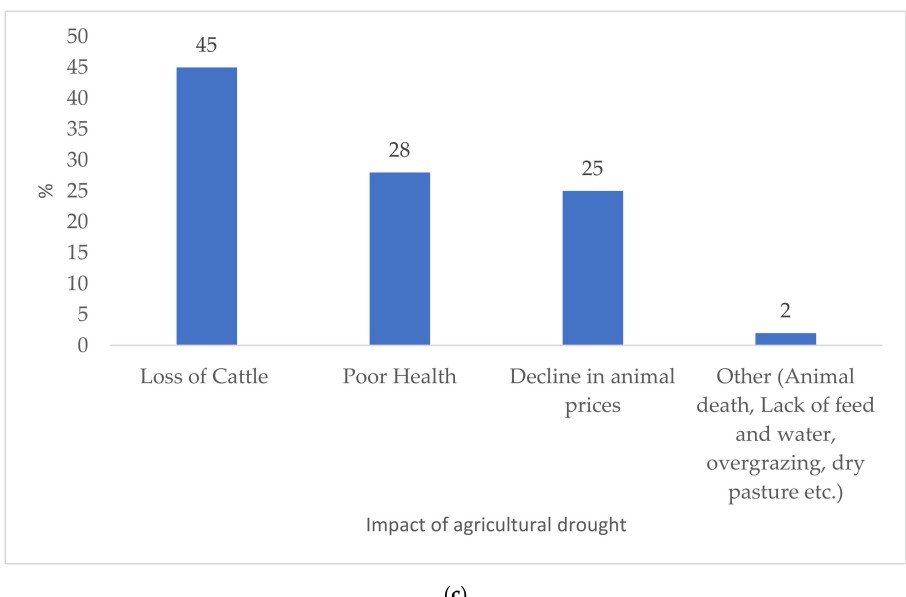

(**c**)

**Figure 4.** Impact of agricultural drought on livestock: (**a**) goats; (**b**) sheep; and (**c**) cattle.

*4.4. Impact on Resilience*

The ADRI of the study area was calculated using Equations (1) and (2). An ADRI greater than zero represents agricultural drought-resilient households while an ADRI less than zero represents households that are not resilient to agricultural drought. The ADRI estimated that 79% (172) of livestock farming households were not resilient to agricultural drought while the remaining 21% (45) were resilient. Agricultural drought has a significant impact on smallholder livestock farmers despite the Northern Cape being one of South Africa's frequently declared drought zones, and any lack or delay in rainfall is likely to result in a drop in food production, including livestock.

*4.5. Impact on the Farm Economy*

4.5.1. Impact of Land

Figure 5 demonstrates the land tenure of livestock farming households per municipality. Respondents from Phokwane (11.06%), Sol Plaatje (14.28%), Dikgatlong (18.89%), and Magareng (6.91%) indicated the use of communal land. Respondents used rented land in Phokwane (0.46%), Sol Plaatje (11.06%), Dikgatlong (7.37%), and Magareng (5.99%). Respondents also indicated customary use of land in Phokwane (3.23%), Sol Plaatje (3.23%), Dikgatlong (11.06%), and Magareng (1.38%). The majority of the respondents indicated that they used communal land for farming, which belonged to the local government. They did not have access or full property rights because they did not own the land, and as a result, they had difficulty obtaining credit. This result implies that one of the reasons for smallholder livestock farmers' low resilience is a lack of access to land. Furthermore, the lack of land ownership exacerbated the severity of the drought's impact on smallholder livestock farmers.

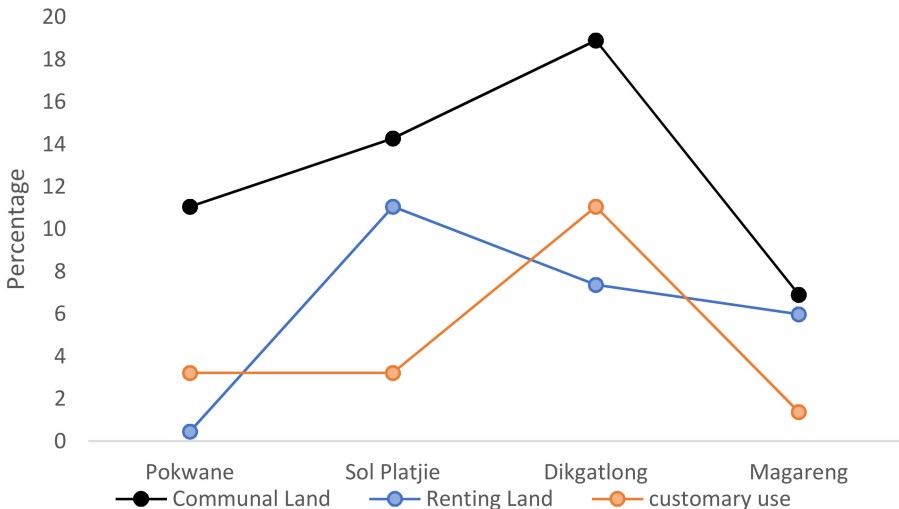

**Figure 5.** Land tenure for smallholder livestock farmers.

### 4.5.2. Impact of Water

Water is a major determinant of agricultural productivity, playing an important role in food security, and aiding drought resilience in agriculture. Boreholes (49%) were the primary source of water on most farms, as seen in Figure 6. Taps or canals were used as alternative water sources (30%). The rest of the respondents relied on wells (18%), lakes (1%), and none (2%), which meant they obtained their water directly from a river. Despite respondents' access to boreholes, not all boreholes were operational. Respondents stated that they occasionally had to transport water from their homes to the farm, resulting in additional production costs (petrol used for transporting the water).

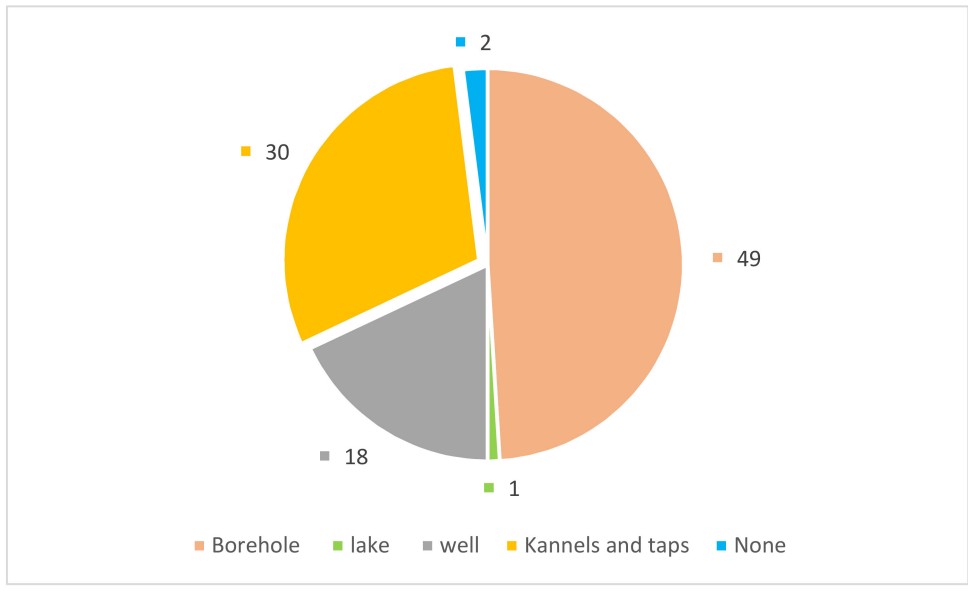

**Figure 6.** Smallholder livestock farmers' water sources.

### 4.5.3. Impact of Assets

Table 4 presents household asset ownership. The majority of the households owned houses (90.8%); farmhouses are appreciating farm assets, and therefore contributed to the farm economy. This was followed by ownership of wheelbarrows (63.1%). Tractors are one of the significant assets that farmers can own to increase production. However, the study showed that tractors were the least owned asset (11.1%), the reason being that the

study focused on livestock rather than crop farmers. The impact of drought on ownership of an asset is significant; the severe drought forced smallholder farmers to sell some of their belongings to cope with the drought. This implies that a household with more assets may have been able to improve their resilience. This also lends support to the hypothesis that the more assets a household possesses, the higher the level of resilience, and the greater adaptive capacity a household possesses, the higher the level of resilience.

**Table 4.** Household asset ownership.

| Variable | Percentage | Frequency |
|---|---|---|
| House | 90.8 | 197 |
| Car | 38.7 | 84 |
| Tractor | 11.1 | 24 |
| Wheelbarrow | 63.1 | 137 |
| Feeding equipment | 36.4 | 79 |
| Livestock trailer | 12.9 | 28 |
| Water tank | 34.1 | 74 |
| Corral system | 18.4 | 40 |
| Computer | 15.2 | 33 |

Source: Author's observation.

*4.6. Impact of Household Economy*

4.6.1. Impact of Food Security

Changes in rainfall patterns contribute to food insecurity in a variety of ways. These include a decrease in agricultural production, household income, expenditure, and job opportunities, and restricted access to credit due to a lack of collateral and economic resources, and higher food prices. Food security is directly or indirectly related to consumption, employment, credit, and saving. Food insecurity was measured using the HFIAS. Food insecurity was experienced differently among respondents, depending on their spending power and the level of financial well-being. Most of the respondents (71%) worried about not having food, 62.7% ate limited amounts of food, and 60.4% ate smaller meals than they felt were needed. More than half of the respondents (56.2%) ate fewer meals in a day, 55.3% did not eat what they preferred, and 55.3% ate what they did not want to eat. Less than half of the respondents (42.4%) experienced not having food in the household, 36.4% experienced going to sleep without food, and 34.1% reported going the whole day without eating. Food-secure households had higher resilience to food insecurity, whereas severely food-insecure households had lower resilience to food insecurity (Table 5).

**Table 5.** Household Food Insecurity Access Scale (HFAIS).

| HFIAS | Response (%) | | Frequency (%) | | |
|---|---|---|---|---|---|
| | | | Rarely | Sometimes | Often |
| Worry about not having food | No | 29 | | | |
| | Yes | 71 | 28.62 | 24.42 | 17.96 |
| Not eat when you prefer | No | 44.70 | | | |
| | Yes | 55.30 | 18.91 | 24.89 | 11.50 |
| Eat limited food | No | 37.30 | | | |
| | Yes | 62.70 | 20.31 | 29.47 | 12.92 |
| Eat what you do not want | No | 44.70 | | | |
| | Yes | 55.30 | 27.65 | 19.80 | 7.85 |
| Eat a smaller meal than felt was needed | No | 39.60 | | | |
| | Yes | 60.40 | 21.20 | 26.27 | 12.93 |
| Eat meals in a day | No | 57.60 | | | |
| | Yes | 42.40 | 19.38 | 16.58 | 6.44 |
| Go to sleep without food | No | 63.60 | | | |
| | Yes | 36.40 | 18.42 | 11.98 | 6 |
| Go the whole day without eating | No | 65.90 | | | |
| | Yes | 34.10 | 18.45 | 13.81 | 1.84 |

Source: Author's compilation based on the survey (2021).

Figure 7 shows the resilience score, which is further explained in Table 6. High scores, over time, indicate that the household concerned is becoming increasingly resilient. Stable

scores, over time, suggest that the situation in the household remains unchanged and that they might require additional services and support. Low scores, over time, indicate that the household is becoming more vulnerable and needs additional services and support.

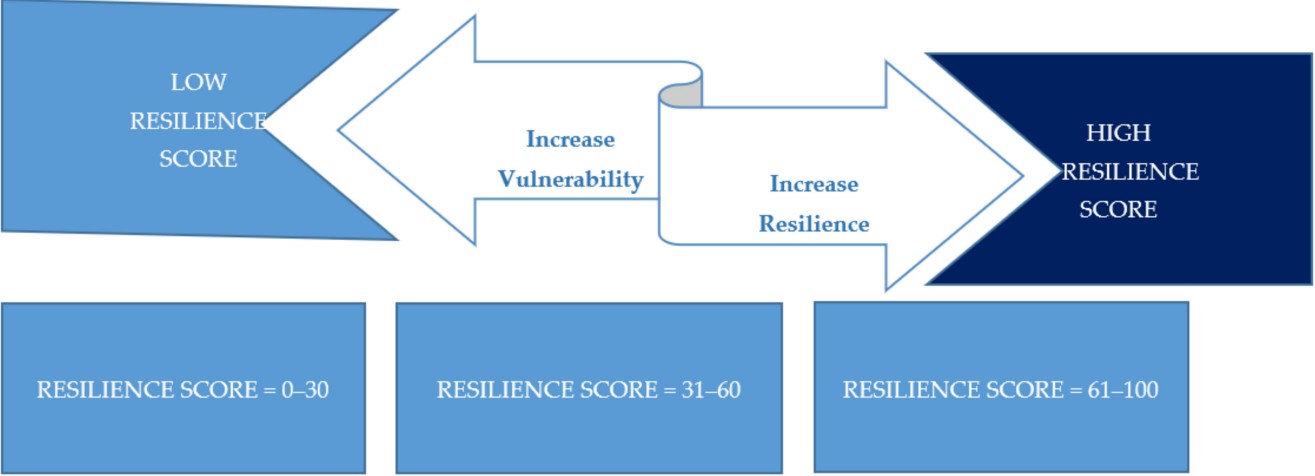

**Figure 7.** Interpreting resilience scores.

**Table 6.** Household resilience levels to food insecurity.

| Categories of Food Insecurity | Frequency (Score) | | |
|---|---|---|---|
| | Rarely/Strong | Sometimes/Moderate | Often/Weak |
| Food secure | 61 | 53 | 36 |
| Mildly insecure | 59 | 42 | 17 |
| Moderately food insecure | 43 | 56 | 23 |
| Severely insecure | 40 | 29 | 4 |

Source: Author' compilation, based on the Survey.

Table 6 shows the household resilience to food insecurity. Food-secure households were associated with stronger resilience to food insecurity since those households experienced none of the food insecurity conditions or only worried (although rarely). In comparison, severely food-insecure households were associated with weaker resilience to food insecurity. Such weaker resilience was indicated if 1 of the following 3 cases were experienced at least once in the last 4 weeks (last 30 days): running out of food, going to bed hungry, or going a full day and night without eating. As illustrated in Figure 6, the resilience score of 61 showed high and increasing resilience, and the household was food secure. The resilience score of 4 showed low resilience, which increased vulnerability, and the household was severely food insecure, as represented by Table 6.

4.6.2. Impact of Consumption Expenditure

This study found low levels of consumption and expenditure in the study area. Table 7 illustrates the consumption comparison in a normal and drought year and livestock prices. The results implied that the respondents in the FBDM experienced low consumption and expenditure levels. For example, the 5th percentile price for livestock was R00.00. Households in a normal year appeared to consume more than in a drought year, and none of the households reported zero consumption. These comparisons suggested that consumption in a normal year exceeded consumption in a drought year by more than 36% at the 5th percentile, and 24% at the 10th percentile. Average consumption in a normal year was over 1.55 times the average consumption in a drought year at 34.3959/22.2301 kg.

**Table 7.** Distribution of expenditure and consumption of respondents in the FBDM.

| | Percentiles | | | | | | |
|---|---|---|---|---|---|---|---|
| | **5th** | **10th** | **25th** | **50th** | **75th** | **90th** | **95th** |
| Consumption in normal year | 3.75 | 5.00 | 5.00 | 10.00 | 20.00 | 55.00 | 162.50 |
| Consumption in drought year | 1.73 | 3.00 | 5.00 | 7.66 | 15.00 | 50.00 | 112.50 |
| Livestock price bought in 2019 | 0.00 | 2.50 | 1200.00 | 2350.00 | 7400.00 | 12,050.00 | 17,750.00 |

Source: Author' compilation, based on the survey.

### 4.6.3. Impact of Employment

The drought's potential impact on employment should not be underestimated. Employment status is essential when measuring the drought resilience of households. Figure 8 shows that 62.2% of the household heads were full-time farmers. This reflected farmers' high level of dependency on production. Few household heads (14.7%) were formally employed, unemployed (13.4%), or pensioners (9.7%).

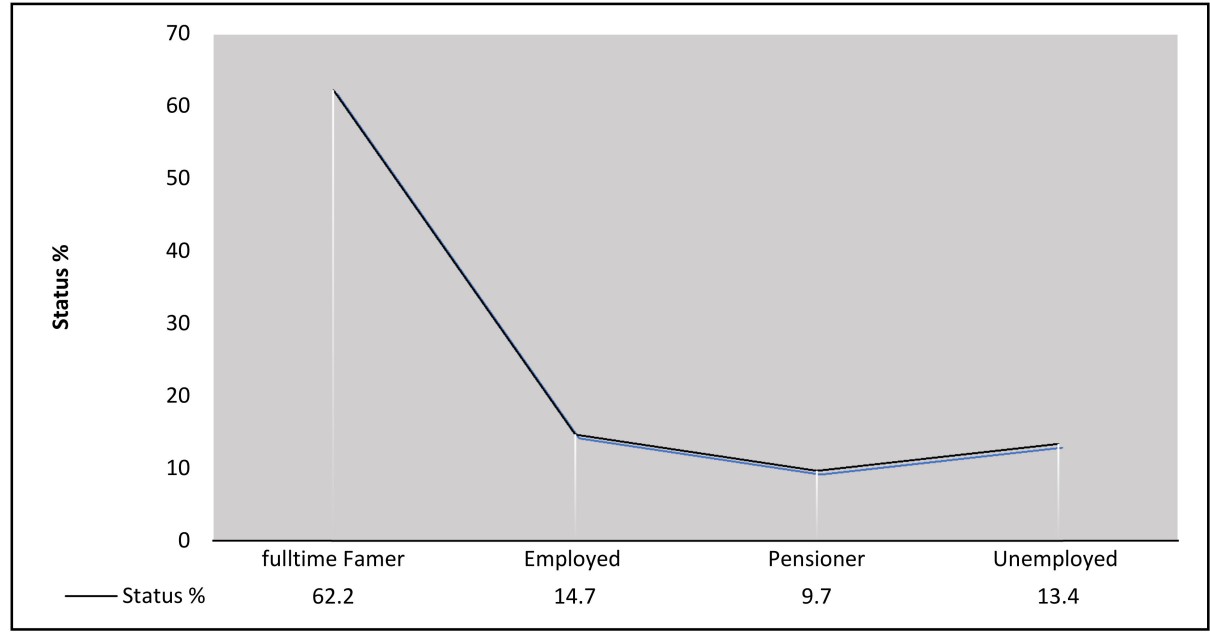

**Figure 8.** Employment status of household heads.

### 4.6.4. Impact of Credit and Saving

Access to finance reduces the vulnerability of smallholder farmers and increases their resilience. Smallholder farmers lacked access to any form of credit as seen in the results, where only 11% of the respondents had access to credit while the remaining 89% did not. Savings act as a stimulus for capital formation and a driver for economic development and progress. The results showed that only 9.2% of the respondents had savings while 90.8% did not.

### 4.7. Impact on Government Policy

There are difficulties associated with the government's relief program. The South African government's relief program includes feed for livestock (fodder), medication for livestock, improving access to agricultural credit and farm input, and increasing smallholder farmers' participation in agricultural drought resilience activities. Resilience activities, such as training and dissemination of information, are among the government's relief measures. However, this study found that government relief during drought was limited and was not disseminated on time.

## 5. Discussion

When it comes to farming, age is a debatable topic. The farmers' average age was 52 years, which is higher than the South African national median age of 24 years [52]. It is concerning that the average age of farmers was relatively high, which means that fewer young people are farming and have joined other industries. This could be due to a lack of funding for start-up farmers and the negative stigmas surrounding agriculture as a career choice. This finding is supported by Meterlerkamp et al. [53], who found that one-third of young people had a positive attitude towards farming and chose agriculture as a career.

The male household heads spent more years in school than their female counterparts. This implies that the more educated and higher-skilled household heads are likely to be the least vulnerable to climate shocks, such as agricultural drought. This is consistent with the finding of Brenda [54], who highlighted that the more educated and higher-skilled individuals of a household are more likely than lesser skilled individuals to be resilient to climate shocks, such as agricultural drought, because they can obtain information about climate change to assess their situation. Almost three-quarters and more than half of the respondents were men and married, respectively. These results imply that men are more involved in livestock farming than women. Furthermore, this study found that married people are more engaged in agriculture, possibly because married household heads can make better decisions during agricultural drought with the assistance of their partners. This finding is in line with a study by Ngeywo et al. [55], who found that youth who dedicate their energy to farming as a business were denied a chance to do so because they believed they were not responsible enough if they were not married.

Agricultural drought is the leading cause of livestock loss. As shown in Figure 3, a large portion of the respondents' livestock (sheep, goats, and cattle) died due to the agricultural drought or suffered ill health as a result of a lack of medication. Even if farmers were willing to sell to avoid loss, livestock prices plummeted due to the agricultural drought. These finding agreed with Vetter et al. [23], who analysed goat and cattle census data from the Msinga area of KwaZulu-Natal, representing the livestock of approximately 3000 households. They found that cattle farmers in Msinga lost 43% of their herd while goat farmers lost 29% of their goats. Drought has a substantial impact on average herd size, livestock feeding, and sheep flock [9]. Furthermore, the findings of this study agree with those of Agri SA [56]. Agri SA reported that the drought had a severe and negative impact on the livestock industry (cattle, sheep, and goats), with the national herd decreasing by 15% and the drought causing a 23% increase in slaughtered cattle and a 37% increase in sheep slaughtered sheep.

According to the ADRI, more than three-quarters of smallholder livestock farmers are not resilient to agricultural drought. This suggests that smallholder livestock farmers need assistance from the government and different stakeholders in industry to enhance their resilience. Smallholder farmers' participation in agricultural drought resilience activities can be improved by providing training and disseminating information, providing fodder and medication for livestock, and improving access to agricultural credit and farm input. This conclusion is consistent with Matlou and Bahta's [40], who found that most farming households in the Northern Cape were not drought resilient. Furthermore, these findings align with those of Boukary et al. [57], who found that a climate change indicator approximated by a lack of rainfall (drought) had a negative and significant effect on rural households' resilience.

The land tenure of livestock farming households is very low. As a result, the lack of land ownership increased the severity of the drought's impact on smallholder livestock farmers. Natural grazing is essential for livestock production, which obviously necessitates more land. Government intervention to assist smallholder farmers, allowing them to grow, could be one solution. According to the World Farmers' Organization [58], many smallholder farmers work on land they do not own, exacerbating their poverty. This is attributed to a lack of political power and equal recognition of basic rights.

Droughts have serious consequences regarding the availability of water. Boreholes and taps were the primary sources of water on the majority of farms in the study area. Water scarcity was exacerbated by agricultural drought, and a variety of factors determined vulnerability. This study's findings agree with those of Peña-Guerrero et al. [59], who found that drought seriously threatened Chile's agriculture, including the livestock sector. The differential vulnerabilities of communities are based on socio-demographics, water security, natural disasters, and climate variability [60].

The more assets a household possesses, the more resilient the household is to agricultural drought. Assets are an important component of household resilience [57]. However, this study showed that the majority of the households owned houses (farmhouses). Drought significantly impacted asset ownership and the severe drought forced smallholder farmers to sell their belongings to cope with the drought. Similarly, Bahta [61] discovered that the majority of smallholder farmers sell their assets as a coping strategy.

Respondents experienced food insecurity differently depending on their spending power and level of financial well-being. The majority of respondents were concerned about running out of food supplies. The resilience score of 61 indicated that these households were resilient, growing, and food secure. A score of 4 indicated low resilience, which increased vulnerability, and that the households were severely food insecure. Hussein et al. [62] reported similar results, where households experienced uncertainty related to food availability due to agricultural drought. Further, Chai et al. [63] highlighted that low scores over time indicated that households were becoming more vulnerable and needed additional services and support. The findings of this study were consistent with the results of Chai et al. [63], where high scores indicated that a household was becoming increasingly resilient, and low scores indicated that a household was becoming more vulnerable and required additional services and support.

Regarding consumption expenditure, households in a normal year appeared to consume more than in a drought year. The FAO [64] noted that prolonged periods of drought, particularly at the end of 2016, were critical constraints on meeting the increasing demand for meat consumption in South Africa since 2015. The number of animals available for slaughter was reduced substantially, leading to a decline in meat consumption of 11.9% to 1 million tonnes, the lowest since 2013.

Most smallholder farmers were full-time farmers, with 9.7% being pensioners. The findings, consistent with Matlou and Bahta [40], suggest that formally employed farming family heads may be more robust to agricultural drought since they have access to finances (for example, personal loans) and can use their monthly salaries. This result implies that the lack of diversification in farm activities reduces the resilience of smallholder farmers. Further, Gray et al. [65] highlighted that agricultural drought decreases the likelihood of employing an individual by approximately 3.2 percentage points. According to Agri-SA [41], the number of farms affected in South Africa's Northern Cape increased from 10,000 to 15,500, covering more than 20 million hectares. This applies to 613,447 livestock units. The total area affected by the drought in the province was 27 million hectares, with a carrying capacity of over 1 million livestock units. The production value in the affected area is estimated to be R 2.53 billion per year. As a result of this, direct reliance on agriculture employment decreased by 22.5%.

Smallholder farmers had no access to credit, and the majority had no savings. This implied that the lack of access to credit and their inability to borrow during the severe drought worsened the smallholder livestock farmers' conditions. This was not surprising given that most livestock farmers lacked access to credit, especially during a drought. Similarly, Matlou and Bahta [40] discovered that only 9% of the smallholder livestock farmers in their study were resilient to agricultural drought. They found that farming households with access to credit were more resilient to agricultural drought than those who did not have access. These findings agree with the International Finance Corporation [66], who found that the percentage of smallholders with access to finance was difficult to quantify. According to estimates, even promising approaches to expanding smallholder

lending, such as value chain finance, reached fewer than 10% of smallholders. Besides, the savings of smallholders are not enough to act as a buffer during drought. The findings of Wieliczko et al. [67] align with this study, proving that the level of savings, especially smallholder farmers, is low, thus limiting their resilience.

Ton et al. [68] highlighted that drought relief for agricultural innovation is common, but drought relief funds that are explicitly for smallholder farmers remain relatively rare. According to this study, government relief during a drought is limited, delayed, or not provided at all. This finding is in line with Matlou and Bahta [40] and Bahta [61], who determined the factors that influence the resilience of smallholder farming households and their coping strategies in the Northern Cape province of South Africa. They found that during the 2015/2016 drought, farmers received assistance from the government (coupons to purchase feed) based on the number of livestock the farmer had. Farmers, however, argued that this was not enough as agricultural drought lasts for a longer period of time and although the government assisted, it was very late; some farmers had already started losing or selling livestock.

Furthermore, the financings are consistent with Ton et al. [68], who reviewed the effectiveness of drought relief support provided by the governments of Malawi, Latin America, Uganda, and Colombia. They found that the drought relief support was insufficient. Furthermore, the little assistance that was received usually arrived too late for the farmers. Mdungela et al. [69] investigated the factors that influence communal farmers' choice regarding coping strategies to sustain productivity during drought in the Eastern Cape province of South Africa. They found that drought relief does not reach farmers in time, and they have to wait for officials from a national department to declare their farm a disaster area.

Ncube [70] sought to characterize smallholder farmers in South Africa's Limpopo and Western Cape provinces, and to identify the coping and adaptation strategies they employ during droughts. The author discovered that there is government provision of fodder for the worst affected livestock farmers, but the processes are inefficient, and often the assistance comes too late. Fan and Rue [71] assessed various challenges that threaten global food security and nutrition. They highlighted that smallholder farmers are a key to ending hunger and undernutrition worldwide, but they do not all receive the same level and kind of government support; they are not a homogenous group.

## 6. Conclusions

The impact of agricultural drought on smallholder livestock farmers was investigated in this study. Data from 217 smallholder livestock farmers from the FBDM in the Northern Cape Province of South Africa were collected and analysed descriptively and the ADRI and HFIAS were used. This study found that most smallholder farmers in the Northern Cape were not resilient to agricultural drought. Additionally, the impact of agricultural drought on resources, food security, and government policy was significant. This implies that smallholder livestock farmers had limited access to land, inadequate water supply, limited access to credit, lack of savings, and lack of diversification. The ineffectiveness of government relief programs during drought hindered respondents' level of production and reduced their resilience to agricultural drought.

This study recommends that improving policy is crucial to enhance the resilience of smallholder livestock farmers. The policy should not be limited to drought relief but should also improve access to land so that smallholder farmers are able to gather more resources and assets. The government needs to work with stakeholders to enhance the resilience of smallholder farmers by supporting less resilient farmers. As a result, these policies can help smallholder farmers be more resilient in times of climatic shock.

In general, this study's findings suggest that governments and non-governmental policymakers should focus on improving the resilience of smallholder farmers by focusing on expanding access to resource bases, reducing food insecurity, and delivering timeous drought relief aid.

This study is unique and adds to existing knowledge as one the first studies to incorporate resources (access to land, access to water, assets), food security (consumption, employment, credit, saving), and government policy (drought relief program) when assessing the effect of agricultural drought on smallholder livestock farmers.

This study used primary data collected through face-to-face interviews to assess the impact of agricultural drought on the resilience of smallholder farming households in the Northern Cape Province. The COVID-19 pandemic caused some data collection delays, and a language barrier was also a limitation. The most widely spoken languages in the Northern Cape are Afrikaans and Setswana (local South African languages), making communication between the researcher and the respondents difficult.

This study recommends that future research in developing countries should investigate the impact of agricultural drought on nutritional security for smallholders and commercial livestock and crop farmers, which was beyond the scope of this study.

**Author Contributions:** All authors significantly contributed to the present manuscript preparation. V.A.M. was involved in data collection, analysis, and writing the first draft. Y.T.B. was a supervisor of the first author, aided in the study design and conceptualization, review, and writing the final draft. All authors have read and agreed to the published version of the manuscript.

**Funding:** National Research Foundation (NRF) of South Africa funded this research, grant number TTK170510230380.

**Data Availability Statement:** Data will be available on request from the corresponding author (Y.T.B.).

**Acknowledgments:** The authors acknowledge and thank the anonymous reviewers for their valuable comments and suggestions. The authors also acknowledge Liesl van der Westhuizen (Science Writer) for language editing this manuscript.

**Conflicts of Interest:** The authors declare no conflict of interest.

## Glossary

| | |
|---|---|
| ADRI | Agricultural Drought Resilience Index |
| AFASA | African Farmers' Association of South Africa |
| Agri SA | Agricultural South Africa-a federation of agricultural organisations, consists of provincial, commodity and corporate members |
| Bartlett's Test of Sphericity | Compares an observed correlation matrix to the identity matrix |
| Chi-square | Is a statistical test used to examine the differences between categorical variables from a random sample in order to judge goodness of fit between expected and observed results. |
| °C | Degree Celsius |
| DAFF | Department of Agriculture, Forestry and Fisheries |
| EPPI | Evidence for Policy and Practice Information |
| FANTA | Food and Nutrition Technical Assistance Project |
| FAO | Food and Agriculture Organization of the United Nations |
| FBDM | Frances Baard District Municipality |
| HFIAS | Household Food insecurity Access Scale |
| IFC | International Finance Corporation |
| IFPRI | International Food Policy Research Institute |
| Km$^2$ | Square kilometre or Kilometre squared |
| KMO | Kaiser–Meyer–Olkin |
| KPMG | Klynveld Peat Marwick Goerdeler-global network of professional firms providing audit, tax and advisory services |
| Md | Number of months household consumed food in a drought year |
| Mn | Number of months household consumed food in a normal year |
| Mm | Millimetre |
| NDAFF | Northern Cape Department of Agriculture, Forestry, and Fisheries |

| NRF | National Research Foundation |
| PCA | Principal Component Analysis |
| Pd | Production of livestock in drought year |
| Pn | Production of livestock in normal year |
| *p*-value | Measure of the probability that an observed difference could have occurred just by random chance |
| Stats SA | Statistics South Africa |
| UNCCD | United Nations Convention to Combat Desertification |
| USD | United States Dollar |
| W | Weight -the loading of components of the first principal weights determined |
| WcnMn | Weight for the number of months during which the household consumed food in a normal year multiplied by actual number of food produced |
| WcdMd | Weight for the number of months during which the household consumed food in a drought year multiplied by actual number of food produced |
| WdPd | Weight of livestock production in normal year multiplied by actual number of livestock produced |
| WnPn | Weight of livestock production in normal year multiplied by actual number of livestock produced |
| WRC | Water Research Commission |

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
