# Peer review of "The Impact of Agricultural Drought on Smallholder Livestock Farmers: Empirical Evidence Insights from Northern Cape, South Africa"

_agriculture, doi:10.3390/agriculture12040442_

Round 1
Reviewer 1 Report
Introduction: 1. The writing order of Introduction should be adjusted. First, the climate and agricultural drought and the impact on animal husbandry should be illustrated. Then came the agricultural drought in the Northern Cape. Finally, a literature review. 2. The content of the literature review is too simple,it need more detailed writing.
Materials and Methods: More information about the study area is required, such as landform, climate, land resources, animal husbandry, farmers' income, etc.
Results:
- Please check the structure of 3.5 and 3.6, corresponding to 3.4.
- The authors simply described the data of the graph in the course of Results. The results of the chart should be analyzed in detail, and the causes and possible consequences should be explained as much as possible.for instance: 3.2:Drought also has a negative impact on animal health, drought led to declines in livestock prices ”. 3.3:the results of the ADRI. 3.4.1:The result implies that the lack of ownership of the land enhances the severity of drought impact on smallholder livestock farmers. 3.4.2:How can drought affect the production and life of farmers by affecting agricultural or domestic water? 3.6:this study found that the government relief during drought is limited; the drought relief does not come on time or late.
Discussion:
- Agricultural drought is an important threat to agriculture (animal husbandry) around the world. We must go deeper into the transfer of results. In addition, a debate should be generated and comparisons made with other regions in South Africa and abroad. In this sense, the literature is excessively South Africa. More references and discussion with the international bibliography are suggested in this part.
- The advantages, shortcomings, innovation points, future research prospects, further discussion of important content, and policy suggestions should also be added in the Discussion.
Author Response
Please find attached response to the reviewer one comments with proof of language editing

Reviewer 2 Report
The manuscript evaluated the impact of agricultural drought on smallholder farming households' resilience in the Northern Cape Province, South Africa. This study found that agricultural drought significantly impacted resources, food security, and government policy. Overall, this study addresses a topic of high relevance for research and also for practice. However, I believe some issues need revision and clarification.

Author Response
Please find attached response to the reviewer two comments with proof of language editing

Round 2
Reviewer 1 Report
The authors have carefully revised the manuscript, which is very good. However, there are still many problems. I think that the Introduction, Conceptual framework and Materials and Methods need major modification.
- Introduction
- To make the expression clearer, the order of some sentences needs to be adjusted. I suggest adjusting the content of 168-176 lines to 34 lines (before In South Africa…) , 44-48 lines(The Northern Cape drought is…) to the end of the first paragraph, and "The impact of agricultural… " to the beginning of the third paragraph (Smallholder livestock farmers…).
- Most content of Literature Review is fragmented information. I suggest that the authors should focus on the research focus of this manuscript. First, the impact of drought on agricultural production, especially animal husbandry production. Second, the recovery of peasant families, especially those engaged in livestock production, after being affected by the drought. Third, how does the government try to alleviate the impact of drought on agriculture or peasant households through policy means. The contents of Literature Review is compiled from these three aspects.
- Conceptual framework
- Repeat “2. Conceptual framework” with “2.1.Conceptual framework”, please delete “2.1. Conceptual framework”.
- The authors simply explained the indicators included by the Conceptual framework. That's not what we want to see, we can know through the pictures and tables. I suggest that the author should explain why this Conceptual framework is built and the interrelationship between these contents.
- Materials and Methods
- The author mentioned “The climate in the Northern Cape is arid and semi-arid. It is a large, dry area with a wide range of temperatures and topographical features. Rainfall is infrequent, ranging from 50 to 400 mm per year. Summer temperatures frequently rise above 40 °C. ” As a reader, I wonder about the geomorphology and the detailed climatic features of this region. Such as in mountains or plains? Average annual precipitation, temperature, and annual evaporation volume? Summer and winter climate conditions? Specific drought conditions, such as water resources, historically suffered from drought. It is clear that the expression of this information is unclear or absent.
- A spatial distribution map of the precipitation should preferably be added to the picture of the study area.
Author Response
Please find attached response to the first reviewer

Reviewer 2 Report
Most of my remarks from the previous round of revision have been addressed. Yet, there are some issues that need to be addressed prior to publication.

Author Response
Please find attached response to the second reviewer
